# FEDPOP: FEDERATED POPULATION-BASED HYPERPARAMETER TUNING

## ABSTRACT

Federated Learning (FL) is a distributed machine learning (ML) paradigm, in which multiple clients collaboratively train ML models without centralizing their local data. Similar to conventional ML pipelines, the client local optimization and server aggregation procedure in FL are sensitive to the hyperparameter (HP) selection. Despite extensive research on tuning HPs for centralized ML, these methods yield suboptimal results when employed in FL. This is mainly because their "training-after-tuning" framework is unsuitable for FL with limited client computation power. While some approaches have been proposed for HP-Tuning in FL, they are limited to the HPs for client local updates. In this work, we propose a novel HP-tuning algorithm, called Federated Population-based Hyperparameter Tuning (FedPop), to address this vital yet challenging problem. FedPop employs population-based evolutionary algorithms to optimize the HPs, which accommodates various HP types at both the client and server sides. Compared with prior tuning methods, FedPop employs an online "tuning-while-training" framework, offering computational efficiency and enabling the exploration of a broader HP search space. Our empirical validation on the common FL benchmarks and complex real-world FL datasets, including full-sized Non-IID ImageNet-1K, demonstrates the effectiveness of the proposed method, which substantially outperforms the concurrent state-of-the-art HP tuning methods in FL.

## 1 INTRODUCTION

Federated Learning (FL) is an effective machine learning paradigm suitable for decentralized data sources (McMahan et al., 2017). Similar to the conventional ML algorithms, FL exhibits sensitivity to empirical choices of hyperparameters (HPs), such as learning rate, and optimization steps (Kairouz et al., 2021). Hyperparameter Tuning (HPT) is a vital yet challenging component of the ML pipeline, which has been extensively studied in the context of centralized ML (Hutter et al., 2019). However, traditional HPT methods, such as Bayesian Optimization (Snoek et al., 2012), are not suitable for FL systems. These methods typically utilize the "training-after-tuning" framework. Within this framework, a substantial number of HPs needs to be evaluated, which involves repetitive training of models until convergence and subsequent retraining after optimizing the optimal HP. Such approaches can drastically increase the client's local computational costs and communication overheads, as it needs to execute multiple federated communications when evaluating only one HP. Furthermore, the distributed validation datasets impose a major challenge for HPT in FL, making it infeasible to evaluate HP for a large number of participating clients.

Recently, a few approaches have emerged to address the problem intersection of HPT and FL, but they still exhibit certain limitations: FedEx (Khodak et al., 2021) pre-defines a narrower HP search space, while FLoRA (Zhou et al., 2021) requires costly retraining after HP-optimization. Moreover, they are only applicable for tuning the client's local HPs. In this paper, we propose Federated Population-based Hyperparameter Tuning (`FedPop`) to address the challenge of tuning HPs for FL. `FedPop` applies population-based evolutionary algorithm (Jaderberg et al., 2017) to optimize the HPs, which adds minimal computational overheads and accommodates various HP types at the client and server sides. Most importantly, `FedPop` employs an online "tuning-while-training" framework, enhancing efficiency and thereby allowing the exploration of a broader HP search space.

In `FedPop`, we first construct multiple HP-configurations as our tuning population, i.e., we initialize multiple tuning processes (members) with randomly initialized HP-configuration, containing the HPs used in the server aggregation and the local client updates. Afterwards, we apply an evolutionary update mechanism to optimize the HPs of each member by leveraging information across different HP-configurations (`FedPop-G`). Hereby, the HPs in underperforming members will be replaced by a perturbed version of the HPs from better-performing ones, enabling an efficient and effective online propagation of the HPs. To further improve the HPs for the local client updates in a fine-grained manner, we consider the active clients in each communication round as our local population, where each member contains one HP-vector used in the local client update (`FedPop-L`). Similarly, evolutionary updates are executed based on the local validation performance of each member to tune these HP-vectors. Most importantly, all the tuning processes, i.e., members of the population, are decentralized and can be asynchronous, aligning perfectly with the distributed system design.

The proposed algorithm `FedPop` achieves new state-of-the-art (SOTA) results on three common FL benchmarks with both vision and language tasks, surpassing the concurrent SOTA HPT method for FL, i.e., FedEx (Khodak et al., 2021). Moreover, we evaluate `FedPop` on large-scale cross-silo FL benchmarks with feature distribution shift (Li et al., 2021), where its promising results demonstrate its applicability to complex real-world FL applications. Most importantly, we demonstrate the scalability of `FedPop`, where we show its applicability to full-sized ImageNet-1K (Deng et al., 2009) with ResNet-50 (He et al., 2016). Our contributions in this paper can be summarized as follows:

- We propose an effective and efficient online hyperparameter tuning (HPT) algorithm, `FedPop`, to address the HPT problem for decentralized ML systems.

- We conduct comprehensive experiments on three common FL benchmarks with both vision and language tasks, in which `FedPop` achieves new SOTA results.

- We verify the maturity of `FedPop` for complex real-world cross-silo FL applications, and further analyze its convergence rate on ImageNet-1K, as well as its effectiveness under different tuning system designs.

## 2 RELATED WORK

**Hyperparameter Tuning for FL System:** Previous works for tuning hyperparameters in FL focus only on specific aspects: Wang et al. (2019) tunes only the local optimization epochs based on the client's resources, while Koskela & Honkela (2018); Mostafa (2019); Reddi et al. (2020) focus on the learning rate of client local training. Dai et al. (2020; 2021) apply Bayesian Optimization (BO) (Snoek et al., 2012) in FL and optimize a personalized model for each client, while Tarzanagh et al. (2022) computes federated hypergradient and applies bilevel optimization. He et al. (2020); Xu et al. (2020); Garg et al. (2020); Seng et al. (2022); Khan et al. (2023) tune architectural hyperparameters, in particular, adapt Neural Architecture Search (NAS) for FL. Zhang et al. (2022) tunes hyperparameter based on the federated system overheads, while Maumela et al. (2022) assumes the training data of each client is globally accessible. Mlodozeniec et al. (2023) partitions both clients and the neural network and tunes only the hyperparameters used in data augmentation. Khodak et al. (2020; 2021) systematically analyze the challenges of hyperparameter tuning in FL and propose FedEx for client local hyperparameters. Zhou et al. (2021) proposes a hyperparameter optimization algorithm that aggregates the client's loss surfaces via single-shot upload. In contrast, the proposed method, FedPop, is applicable to various HP types on the client and server sides. In addition, it does not impose any restrictions on data volume and model architecture.

**Evolutionary Algorithms:** Evolutionary algorithms are inspired by the principles of natural evolution, where stochastic genetic operators, e.g., mutation and selection, are applied to the members of the existing population to improve their survival ability, i.e., quality (Telikani et al., 2021). Evolutionary algorithms have shown their potential to improve machine learning algorithms, including architecture search (Real et al., 2017; Liu et al., 2017), hyperparameter tuning (Jaderberg et al., 2017; Parker-Holder et al., 2020), and Automated Machine Learning (AutoML) (Liang et al., 2019; Real et al., 2020). FedPop employs an online evolutionary algorithm, which is computationally efficient and explores a broader HP search space. To the best of our knowledge, FedPop is the first work combining evolutionary algorithms with HP optimization in Federated Learning.

## 3 FEDERATED HYPERPARAMETER TUNING

### 3.1 PROBLEM DEFINITION

In this section, we introduce the problem setup of hyperparameter tuning for FL. Following the setting introduced by Khodak et al. (2021), we assume that there are $N_c \in \mathbb{N}^+$ clients joining the federated communication. Each client $k$ owns a training, validation, and testing set, denoted by $T^k, V^k$, and $E^k$, respectively. To simulate the communication capacity of a real-world federated system, we presume that there are exactly $K \in \mathbb{N}^+$ active clients joining each communication round. In the classical Fed-

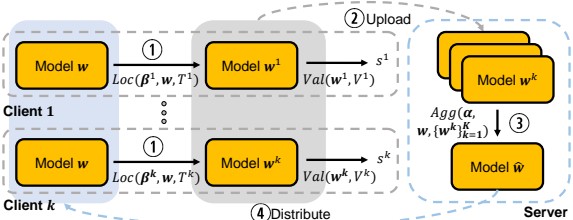

Figure 1: Schematic illustration of the operations involved in one communication round, summarized as Fed-Opt.

Avg approach (McMahan et al., 2017), the central server obtains the model weight $\boldsymbol{w} \in \mathbb{R}^d$ by iteratively distributing $\boldsymbol{w}$ to the active clients and averaging the returned optimized weights, i.e., $\{\boldsymbol{w}^k | 1 \le k \le K\}$.

More specifically, we denote the server aggregation and the client local training functions as Agg and Loc, respectively. Our goal is to tune the hyperparameter vectors (**HP-vectors**) used in these two functions. In particular, we denote the HP-vector used in Agg and Loc as $\boldsymbol{\alpha}$ and $\boldsymbol{\beta}$, which are sampled from the hyperparameter distribution $H_a$ and $H_b$, respectively. We define the combination of $\boldsymbol{\alpha}$ and $\boldsymbol{\beta}$ as one **HP-configuration**. In the following, we explain the general steps executed in the communication round, which involves these functions and HP-configurations. We summarize these steps as federated optimization (Fed-Opt), which is illustrated in Figure 1. Specifically, all active clients first execute function Loc (①) in parallel:

$$\boldsymbol{w}^k \leftarrow \text{Loc}(\boldsymbol{\beta}^k, \boldsymbol{w}, T^k), \tag{1}$$

which takes the HP-vector $\boldsymbol{\beta}^k$, model parameters $\boldsymbol{w}$ distributed by the central server, and the local training set $T^k$ as inputs, and outputs the optimized model weight $\boldsymbol{w}^k$. Afterwards, the central server aggregates $\boldsymbol{w}^k$, uploaded by the active clients (②) , and executes function Agg (③):

$$\hat{\boldsymbol{w}} \leftarrow \text{Agg}(\boldsymbol{\alpha}, \boldsymbol{w}, \{\boldsymbol{w}^k | 1 \le k \le K\}), \tag{2}$$

which takes HP-vector $\boldsymbol{\alpha}$, current model parameter $\boldsymbol{w}$, updated model parameters from the active clients $\{\boldsymbol{w}^k | 1 \le k \le K\}$, and outputs the aggregated model weight $\hat{\boldsymbol{w}}$ which will be distributed to the active clients in the next communication round (④). The goal of the federated hyperparameter tuning method is to find the optimal HP-vectors $\boldsymbol{\alpha}$ and $\boldsymbol{\beta}$ within a predefined communication budget.

### 3.2 CHALLENGES

Given the problem defined in the previous section, we describe the two main challenges when tuning the hyperparameters for federated learning:

**(C1) Extrem resource limitations**: The communication budgets for optimizing ML models via FL are always very constrained due to the limited computational power of the clients and connection capacity of the overall system (Li et al., 2020). Therefore, common hyperparameter tuning algorithms, such as extensive local hyperparameter tuning for each client, or experimenting multiple hyperparameter configurations for the overall federated system and then retraining, may not be suitable in the context of FL.

**(C2) Distributed validation data**: In centralized ML, most hyperparameter tuning algorithms select the HP-configurations based on their validation performance. However, the validation data ($V^k$) is distributed across the clients in FL. Computing a validation score over all clients is extremely costly and thus infeasible for FL. The alternative is to use the validation performance of client subsets, e.g., the active clients of the communication round, which greatly reduces computational costs. However, this may lead to evaluation bias when the distributed client data are not independent and identically distributed (*Non-IID*).

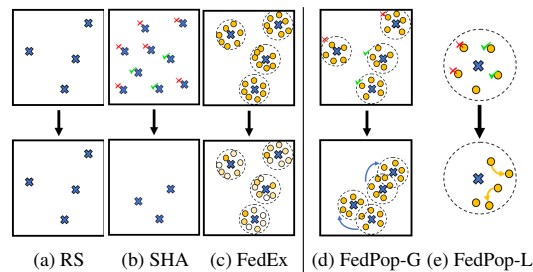

| Method | Number of tried $\alpha$ | Number of tried $\beta$ | Optim. of $\alpha$ | Optim. of $\beta$ |
|--------|------------------------|-------------------------|--------------------|-------------------|
| RS | 5 | 5 | ✗ | ✗ |
| SHA | 27 | 27 | ✗ | ✗ |
| FedEx | 5 | 135 | ✗ | ✓ |
| FedPop | 45 | >1000 | ✓ | ✓ |

$\alpha$: server HP-vector    $\beta$: client HP-vector

✖: HP-configuration (combination of $\alpha, \beta$)

○: additional client HP-vector ($\beta$)

(a) RS    (b) SHA    (c) FedEx  |  (d) FedPop-G (e) FedPop-L

Figure 2: Schematic (*left*) and numeric (*right*) comparison between FedPop and other baselines. (*left*) One *blue cross* represents one HP-configuration, while one *yellow dot* represents an additional client HP-vector used in FedEx and FedPop. FedEx optimizes the sampling probabilities of $\beta$ based on validation performance. In contrast, our method supports the optimization of both **server** (FedPop-G) and **client** (FedPop-G and -L) HP-vectors. (*right*) Number of HP-vectors tested in different HP-tuning methods on CIFAR-10 benchmark. Detailed computation of the numbers is provided in the Appendix. FedPop explores broader search space with the help of evolutionary updates and experiments the largest number of HP-configurations among all methods.

## 3.3 BASELINES

Before introducing the proposed algorithm (FedPop) which addresses the challenges of HP-tuning in FL, we illustrate the adaptation of two widely adopted HP-tuning baselines for FL applications and the notations. For the FL setup, we define the total communication budget and the maximum resources per HP-configuration as $R_t$ and $R_c$, respectively. We devise two baseline methods for tuning $\alpha, \beta$:

(1) **Random Search (RS)** first initializes $N_c(= \frac{R_t}{R_c})$ HP-configurations. Afterwards, an ML model and $N_c$ tuning processes will be initialized, where each tuning process executes $R_c$ federated communication rounds to optimize the model using one HP-configuration. Finally, the optimized models from all tuning processes will be evaluated and the HP-configuration with the best performance is saved.

(2) **Successive Halving (SHA)** is a variation of RS which eliminates $\frac{1}{\eta}$-quantile of the under-performing HP-configurations after specific numbers of communication rounds. Within the same tuning budget $R_t$, SHA is able to experiment more HP-configurations compared with RS, increasing the likelihood of achieving better results. The number of HP-configurations in SHA, $N_c(> \frac{R_t}{R_c})$, is based on $R_t, R_c$ and the number of elimination operations. However, the elimination might also discard HP-configurations which lead to promising results but perform poorly at early stages.

**Limitations:** These baseline methods exhibit two limitations when adapted to FL applications: First, as shown in Figure 2 left, their numbers of HP-configurations, as well as the HP values, are pre-defined and remain fixed throughout the tuning process. Second, these baseline methods are "static" and no active tuning is executed inside each tuning process. Specifically, the model evaluation results are only obtained and utilized after $R_c$ communication rounds. Therefore, we propose FedPop, a population-based tuning algorithm that updates the HP-configurations via evolutionary update algorithms. As a result of its high efficiency, it experiments the largest number of HP-vectors among all methods (Figure 2 right). We introduce FedPop in the following section.

## 3.4 PROPOSED METHOD

The proposed method, Federated Population-Based Hyperparameter Tuning (FedPop), adopts the aforementioned baselines to construct the populations. In the following, we use RS for constructing the initial population of HP-configurations. However, other methods such as SHA can also be applied as a population constructor and we provide detailed explanations in the Appendix.

As shown in Figure 3, we first randomly sample the HP-vectors ($\alpha$ and $\beta$) for each tuning process *in parallel* and execute federated optimization Fed-Opt (Figure 1). Afterwards, we conduct FedPop based on the validation scores $s$ returned from the active clients in each tuning process. FedPop can be divided into 2 sub-procedures: FedPop-L focuses on a fine-grained search of HP-vector $\beta$ inside each HP-configurations (*intra-config*), while FedPop-G aims at tuning both HP-vectors

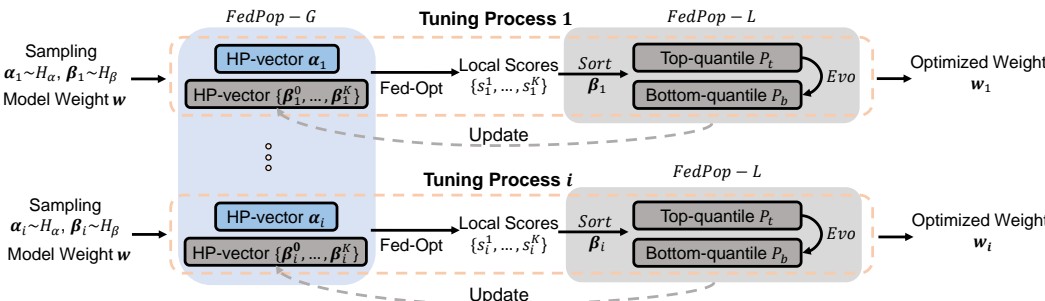

Figure 3: Schematic illustration of `FedPop`, including `FedPop-L` for intra-configuration HP-tuning and `FedPop-G` for inter-configuration HP-tuning. `FedPop` employs an online "tuning-while-training" schema for tuning both server ($\boldsymbol{\alpha}$) and clients ($\boldsymbol{\beta}$) HP-vectors. All functions in `FedPop` can be executed in a parallel and asynchronous manner.

$\boldsymbol{\alpha}$ and $\boldsymbol{\beta}$ across all HP-configurations (*inter-config*). The pseudo codes of the proposed method are given in Algorithm 1.

With `RS` as the population constructor, `FedPop` first randomly initializes $N_c$ HP-configurations, indicated by $(\boldsymbol{\alpha}_i, \boldsymbol{\beta}_i^0)$, and copies the model weight vector $\boldsymbol{w}$. Afterwards, we randomly sample addition $K$ HP-vectors, i.e., $\{\boldsymbol{\beta}_i^k | 1 \le k \le K\}$, inside a small $\Delta$-ball centered by $\boldsymbol{\beta}_i^0$. $\Delta$ is selected based on the distribution of the HP and more details are provided in the Appendix. This is because we find that using too distinct HP-vectors for the active clients would lead to unstable performance, which was also observed by Khodak et al. (2021). We also provide a schematic illustration in Figure 2, where the *yellow dots* ($\{\boldsymbol{\beta}_i^k | 1 \le k \le K\}$) are enforced to lie near the *blue crosses* ($\boldsymbol{\beta}_i^0$). Note that this resampling process of $\boldsymbol{\beta}_i^k$ is also executed when $\boldsymbol{\beta}_i^0$ is perturbed via `Evo` in `FedPop-G`. Finally, $R_c$ communication rounds are executed for each tuning process in parallel, where the validation scores $s_i^k$, of the $k_{th}$ active client in the $i_{th}$ tuning process is recorded.

### 3.4.1 EVOLUTION-BASED HYPERPARAMETER UPDATE (Evo)

Inspired by Population-based Training (Jaderberg et al., 2017), we design our evolution-based hyperparameter update function `Evo` as the following,

$$\text{Evo}(\boldsymbol{h}) = \begin{cases} \hat{h}_j \sim U(h_j - \delta_j, h_j + \delta_j) & \text{s.t.} \quad H_j = U(a_j, b_j), \\ \hat{h}_j \sim U\{x_j^{i \pm \lfloor \delta_j \rceil}, x_j^i\} & \text{s.t.} \begin{cases} H_j = U\{x_j^0, ..., x_j^n\}, \\ h_j = x_j^i, \end{cases} \end{cases} \tag{3}$$

where $\boldsymbol{h}$ represents one HP-vector, i.e., $\boldsymbol{\alpha}$ or $\boldsymbol{\beta}$ for our problem setting. We perturb the $j_{th}$ value of $\boldsymbol{h}$, $h_j$, based on its original sampling distribution $H_j$: (1) If $h_j$ is sampled from a continuous uniform distribution $H_j = U(a_j, b_j)$ (e.g., log-space of learning-rate, dropout), then we perturb $h_j$ by resampling it from $U(h_j - \delta_j, h_j + \delta_j)$, where $\delta_j \leftarrow (b_j - a_j)\epsilon$ and $\epsilon$ is the pre-defined perturbation intensity. (2) If $h_j = x_j^i$ is sampled from a discrete distribution $H_j = U\{x_j^0, ..., x_j^n\}$ (e.g., batch-size, epochs), then we perturb $h_j$ by reselecting its value from $\{x_j^{i-\lfloor \delta_j \rceil}, x_j^i, x_j^{i+\lfloor \delta_j \rceil}\}$. To further increase the diversity of the HP search space during tuning, we resample $h_j$ from its original distribution $H_j$ with the probability of $p_{re}$. While the HPs are randomly initialized in the early tuning stages, they become more informative as training progresses. To reflect this in `FedPop`, we employ a cosine annealing schema to control the values of $\epsilon$ and $p_{re}$ based on the conducted communication rounds. More details are provided in the Appendix.

### 3.4.2 FEDPOP-G FOR INTER-CONFIGURATION TUNING

In `FedPop-G`, we adopt the average validation loss of all active clients, i.e., $s_i = \frac{1}{K} \sum_{k=1}^{K} s_i^k$, as the performance score for $i_{th}$ HP-configuration. However, $s_i$ may be a biased performance measurement, i.e., the disparity in the difficulty of the validation sets between different clients may lead to noisy $s_i$. To reduce the impact of the noise, `FedPop-G` is conducted after every $T_g$ communication

rounds. Hereby, the list of scores $s_i$ over $T_g$ rounds are recorded and their weighted sum with a power-law weight decay is utilized as the measurement. The tuning procedure starts by sorting the HP-configurations according to their validation scores. Afterwards, 2 subsets, i.e., $\boldsymbol{Q}_b$ and $\boldsymbol{Q}_t$, are constructed, representing the indices of the bottom and top $\frac{1}{\rho}$-quantile of the HP-configurations, respectively. Finally, the HP-configura-

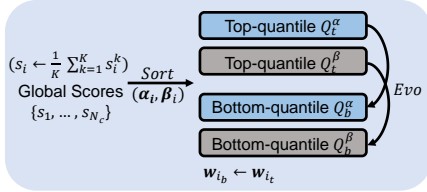

Figure 4: FedPop-G.

tions with indices in $\boldsymbol{Q}_b$ will be replaced by the perturbed version of the HP-configurations with indices in $\boldsymbol{Q}_t$. Specifically, $\boldsymbol{\alpha}_{i_b}, \boldsymbol{\beta}_{i_b}^0$ are replaced by the perturbed version of $\boldsymbol{\alpha}_{i_t}, \boldsymbol{\beta}_{i_t}^0$ via Evo (Equation 3), the model weight in $i_b$-$th$ HP-configuration ($\boldsymbol{w}_{i_b}$) are replaced by the $i_t$-$th$ ($\boldsymbol{w}_{i_t}$).

### 3.4.3 FEDPOP-L FOR INTRA-CONFIGURATION TUNING

To further explore the local neighborhood of $\boldsymbol{\beta}_i^0$ for client local update in a fine-grained manner, we apply FedPop-L inside each tuning process. Hereby, we provide an informative assessment of $\boldsymbol{\beta}_i^0$ and its local neighborhood to enhance the robustness of HP-configuration. For simplicity, we omit $i$ in the following notations. We consider the base HP-vector $\boldsymbol{\beta}^0$ as the perturbation center and restrict the perturbated HP-vector to lie inside a $\Delta$-ball of it, i.e., $||\boldsymbol{\beta}^k - \boldsymbol{\beta}^0||_2 \leq \Delta$. At each communication round, $\boldsymbol{\beta}^k$ will be assigned to Loc of the $k_{th}$ active client, the validation loss of the optimized model $\boldsymbol{w}^k$ will be recorded as the score $s^k$ for HP-vector $\boldsymbol{\beta}^k$. Afterwards, $\{\boldsymbol{\beta}^k\}_{k=1}^K$ will be sorted according to the validation scores and separated into 2 subsets, containing the indices of the bottom ($\boldsymbol{P}_b$) and the top ($\boldsymbol{P}_t$) $\frac{1}{\rho}$-quantile of the $\boldsymbol{\beta}$, respectively. Finally, the HP-vectors $\boldsymbol{\beta}^{k_t}$ with indices in $\boldsymbol{P}_t$ will be perturbed to replace the HP-vectors $\boldsymbol{\beta}^{k_b}$ with indices in $\boldsymbol{P}_b$ via Evo.

---

**Algorithm 1:** Federated Population-Based Hyperparameter Tuning (FedPop).

---

**Input:** Number of active clients per round $K$, number of HP-configurations $N_c$, maximum communication budget for each HP-configuration $R_c$, perturbation interval for FedPop-G $T_g$, model weight $\boldsymbol{w}$, $N_c$ server HP-vectors $\boldsymbol{\alpha} = \{\boldsymbol{\alpha}_1, ..., \boldsymbol{\alpha}_{N_c}\}$, $N_c$ client HP-vectors $\boldsymbol{\beta} = \{\boldsymbol{\beta}_1^0, ..., \boldsymbol{\beta}_{N_c}^0\}$.
Copy the model weights $\boldsymbol{w}_i \leftarrow \boldsymbol{w}$ for all $N_c$ tuning processes.

**for** *comm. round* $r \leftarrow 1$ **to** $R_c$ **do**
 **for** $i \leftarrow 1$ **to** $N_c$ **do**
  // **in parallel**
  **if** $len(\boldsymbol{\beta}_i) == 1$ **then**
   Randomly sample $\{\boldsymbol{\beta}_i^k\}_{k=1}^K$ inside $\Delta$-ball of $\boldsymbol{\beta}_i^0$.
  **for** *Client* $k \leftarrow 1$ **to** $K$ **do**
   // **in parallel**
   $\boldsymbol{w}_i^k \leftarrow$ Loc($\boldsymbol{\beta}_i^k, \boldsymbol{w}_i, T^k$)
   $s_i^k \leftarrow$ Val($\boldsymbol{w}_i^k, V^k$)
  $\boldsymbol{\beta}_i \leftarrow$ FedPop-L ($\boldsymbol{\beta}_i, \{s_i^k\}_{k=1}^K, K$)
  $\boldsymbol{w}_i \leftarrow$ Agg($\boldsymbol{\alpha}_i, \boldsymbol{w}_i, \{\boldsymbol{w}_i^k\}_{k=1}^K$)
  $s_i \leftarrow \frac{1}{K}\sum_{k=1}^K s_i^k$
 **if** $r\%T_g = 0$ **then**
  $\{\boldsymbol{\alpha}_i, \boldsymbol{\beta}_i, \boldsymbol{w}_i\}_{i=1}^{N_c} \leftarrow$ FedPop-G ($\{\boldsymbol{\alpha}_i, \boldsymbol{\beta}_i, \boldsymbol{w}_i, s_i\}_{i=1}^{N_c}, N_c$)
**return** $\{\boldsymbol{w}_i\}_{i=1}^{N_c}$

**Function** *FedPop-L*($\boldsymbol{\beta}, \boldsymbol{s}, K$)
 $\boldsymbol{P}_b \leftarrow \{k : s^k \geq \frac{\rho-1}{\rho}\text{-quantile}(\{s^k\}_{k=1}^K)\}$
 $\boldsymbol{P}_t \leftarrow \{k : s^k \leq \frac{1}{\rho}\text{-quantile}(\{s^k\}_{k=1}^K)\}$
 **for** $k_b \in \boldsymbol{P}_b$ **do**
  Sample $k_t$ from $\boldsymbol{P}_t$.
  Delete $\boldsymbol{\beta}^{k_b}$.
  $\boldsymbol{\beta}^{k_b} \leftarrow$ Evo($\boldsymbol{\beta}^{k_t}$)
 **return** $\boldsymbol{\beta}$

**Function** *FedPop-G*($\boldsymbol{\alpha}, \boldsymbol{\beta}, \boldsymbol{w}, \boldsymbol{s}, N_c$)
 $\boldsymbol{Q}_b \leftarrow \{i : s_i \geq \frac{\rho-1}{\rho}\text{-quantile}(\{s_i\}_{i=1}^{N_c})\}$
 $\boldsymbol{Q}_t \leftarrow \{i : s_i \leq \frac{1}{\rho}\text{-quantile}(\{s_i\}_{i=1}^{N_c})\}$
 **for** $i_b \in \boldsymbol{Q}_b$ **do**
  Sample $i_t$ from $\boldsymbol{Q}_t$.
  Delete $\boldsymbol{\alpha}_{i_b}, \boldsymbol{\beta}_{i_b}, \boldsymbol{w}_{i_b}$.
  $\boldsymbol{\alpha}_{i_b}, \boldsymbol{\beta}_{i_b}^0 \leftarrow$ Evo($\boldsymbol{\alpha}_{i_t}, \boldsymbol{\beta}_{i_t}^0$)
  $\boldsymbol{w}_{i_b} \leftarrow \boldsymbol{w}_{i_t}$
 **return** $\boldsymbol{\alpha}, \boldsymbol{\beta}, \boldsymbol{w}$

---

### 3.4.4 SOLUTIONS TO CHALLENGES

(**C1**) FedPop does not require Bayesian Optimization (Zhou et al., 2021) or gradient-based hyperparameter optimization (Khodak et al., 2021), which saves the communication and computation costs. Besides, FedPop utilizes an *online* evolutionary method (Evo) to update the hyperparameters, i.e., not "training-after-tuning" but "tuning-while-training", which eliminates the need for "retraining" after finding a promising HP-configuration. Note that all procedures in FedPop can be conducted in a parallel and asynchronous manner. (**C2**) FedPop-G is conducted every $T_g$ communication rounds to mitigate the noise depicted in the validation scores of HP-configurations. Besides,

Table 1: Evaluation results of different hyperparameter tuning algorithms on three benchmark datasets. We report the *global* and locally *finetuned* (in the brackets) model performance with format mean$_{\pm std}$ from 5-trial runs using different seeds. The best results are marked in **bold**.

| Pop. Con. | Tuning Algo. | CIFAR-10 | | | FEMNIST | | Shakespeare | |
|---|---|---|---|---|---|---|---|---|
| | | IID | Non-IID $(Dir_{1.0})$ | Non-IID $(Dir_{0.5})$ | IID | Non-IID | IID | Non-IID |
| RS | None | 53.26$_{\pm 8.37}$ (43.02$_{\pm 4.02}$) | 48.92$_{\pm 2.75}$ (35.23$_{\pm 7.46}$) | 47.46$_{\pm 10.38}$ (35.35$_{\pm 9.48}$) | 82.86$_{\pm 1.24}$ (83.76$_{\pm 3.56}$) | 79.06$_{\pm 5.59}$ (83.09$_{\pm 2.64}$) | 33.76$_{\pm 11.27}$ (31.19$_{\pm 10.18}$) | 32.67$_{\pm 12.27}$ (31.32$_{\pm 9.92}$) |
| | FedEx | 60.87$_{\pm 8.09}$ (62.48$_{\pm 11.68}$) | 57.04$_{\pm 5.61}$ (56.93$_{\pm 13.36}$) | 59.74$_{\pm 5.05}$ (58.61$_{\pm 9.22}$) | 82.84$_{\pm 0.80}$ (82.57$_{\pm 3.25}$) | 82.14$_{\pm 1.60}$ (84.03$_{\pm 2.48}$) | 42.68$_{\pm 7.24}$ (41.22$_{\pm 6.34}$) | 44.28$_{\pm 8.78}$ (46.69$_{\pm 7.39}$) |
| | **FedPop** | **66.00**$_{\pm 3.97}$ (**69.54**$_{\pm 3.60}$) | **62.25**$_{\pm 5.03}$ (**61.08**$_{\pm 5.32}$) | **61.27**$_{\pm 5.52}$ (**60.36**$_{\pm 5.62}$) | **84.33**$_{\pm 1.41}$ (**85.99**$_{\pm 1.62}$) | **83.21**$_{\pm 2.08}$ (**85.48**$_{\pm 1.48}$) | **44.30**$_{\pm 3.37}$ (**44.46**$_{\pm 3.53}$) | **47.28**$_{\pm 3.47}$ (**50.25**$_{\pm 3.87}$) |
| SHA | None | 72.08$_{\pm 2.52}$ (72.12$_{\pm 3.48}$) | 54.68$_{\pm 6.25}$ (45.08$_{\pm 5.26}$) | 48.99$_{\pm 8.91}$ (34.07$_{\pm 7.10}$) | 83.81$_{\pm 0.45}$ (85.52$_{\pm 1.63}$) | 80.62$_{\pm 2.88}$ (87.64$_{\pm 0.64}$) | 52.23$_{\pm 2.54}$ (49.06$_{\pm 5.98}$) | 51.68$_{\pm 0.95}$ (48.83$_{\pm 3.12}$) |
| | FedEx | 74.12$_{\pm 1.76}$ (72.58$_{\pm 3.10}$) | 65.06$_{\pm 11.89}$ (57.27$_{\pm 14.88}$) | 56.68$_{\pm 11.02}$ (45.13$_{\pm 17.24}$) | 81.19$_{\pm 3.24}$ (85.69$_{\pm 1.91}$) | 82.76$_{\pm 0.54}$ (86.79$_{\pm 2.89}$) | 51.79$_{\pm 1.25}$ (51.89$_{\pm 1.30}$) | 51.26$_{\pm 2.73}$ (51.01$_{\pm 3.36}$) |
| | **FedPop** | **76.69**$_{\pm 1.02}$ (**74.49**$_{\pm 0.56}$) | **73.50**$_{\pm 2.31}$ (**66.44**$_{\pm 3.67}$) | **69.99**$_{\pm 1.98}$ (**57.31**$_{\pm 3.02}$) | **84.33**$_{\pm 0.57}$ (**86.84**$_{\pm 0.98}$) | **83.26**$_{\pm 0.86}$ (**88.33**$_{\pm 0.79}$) | **53.48**$_{\pm 0.57}$ (**52.66**$_{\pm 1.91}$) | **53.07**$_{\pm 0.97}$ (**52.79**$_{\pm 0.36}$) |

FedPop-L dynamically searches and evaluates the local neighborhood of $\beta^0$, providing a more informative judgment of the HP-configuration.

# 4 EXPERIMENTS AND ANALYSES

We conduct an extensive empirical analysis to investigate the proposed method and its viability. Firstly, we compare FedPop with the SOTA and other baseline methods on three common FL benchmarks following Khodak et al. (2021). Subsequently, we validate our approach by tuning hyperparameters for complex real-world cross-silo FL settings. Besides, we conduct an ablation study on FedPop to demonstrate the importance of its components. Moreover, we present convergence analysis of FedPop and its promising scalability by training ResNets from scratch on full-sized Non-IID ImageNet-1K via FL. Finally, we analyze FedPop under different tuning system designs.

## 4.1 BENCHMARK EXPERIMENTS

### 4.1.1 DATASETS DESCRIPTION

We conduct experiments on three benchmark datasets on both vision and language tasks: (1) *CIFAR-10* (Krizhevsky et al., 2009), which is an image classification dataset containing 10 categories of real-world objects. (2) *FEMNIST* (Caldas et al., 2018), which includes gray-scale images of hand-written digits and English letters, producing a 62-way classification task. (3) *shakespeare* (Caldas et al., 2018) is a next-character prediction task and comprises sentences from Shakespeare's Dialogues.

We investigate 2 different partitions of the datasets: (1) For i.i.d (*IID*) setting, we randomly shuffle the dataset and evenly distribute the data to each client. (2) For non-i.i.d (*Non-IID*) settings, we follow Khodak et al. (2021); Caldas et al. (2018) and assume each client contains data from a specific writer in FEMNIST, or it represents an actor in Shakespeare. For CIFAR-10 dataset, we follow prior arts (Zhu et al., 2021; Lin et al., 2020) to model Non-IID label distributions using Dirichlet distribution $Dir_x$, in which a smaller $x$ indicates higher data heterogeneity. We set the communication budget $(R_t, R_c)$ to $(4000, 800)$ for CIFAR-10 and shakespeare, while $(2000, 200)$ for FEMNIST following previous works (Khodak et al., 2021; Caldas et al., 2018). For the coefficients used in FedPop, we set the initial perturbation intensity $\epsilon$ to 0.1, the initial resampling probability $p_{re}$ to 0.1, and the quantile coefficient $\rho$ to 3. The perturbation interval $T_g$ for FedPop-G is set to $0.05R_c$. Following Khodak et al. (2021), we define $\boldsymbol{\alpha} \in \mathbb{R}^3$ and $\boldsymbol{\beta} \in \mathbb{R}^7$, i.e., we tune learning rate, scheduler, and momentum for server-side aggregation (Agg), and learning rate, scheduler, momentum, weight-decay, the number of local epochs, batch-size, and dropout rate for local clients updates (Loc), respectively. More details about the search space and the model architectures are provided in Appendix.

### 4.1.2 RESULTS AND DISCUSSION

In Table 1, we report the testing accuracy achieved by the final model after performing hyperparameter tuning with different algorithms on three benchmarks. Hereby, we report the results of the *global* model, which is the server model $w$ after the execution of the final communication round, and the *finetuned* model, which is the final global model finetuned on clients local data via $\text{Loc}(\beta^0, w, T^k)$. We observe that FedPop, combined with either RS or SHA as a population constructor, outperforms all the competitors on all benchmarks. For IID settings, the global model tuned on CIFAR-10 with FedPop, with RS or SHA as a population constructor, outperforms FedEx by 5.13% and 2.57%, respectively. Likewise, FedPop yields the highest average accuracy on FEMNIST and Shakespeare. For Non-IID settings, FedPop achieves a significant improvement of around 3% and 10% on average compared with FedEx in CIFAR-10, when combined with RS and SHA, respectively. Moreover, we find that the performance improvement of the finetuned model (in the brackets) tuned by FedPop surpasses the other baselines. Additionally, we observe that during the tuning procedures, certain trials in the baselines and FedEx fail to converge. We attribute this to their pre-defined and fixed hyperparameters search spaces and values, resulting in higher sensitivity to the hyperparameter initialization. This phenomenon is observed via their larger accuracy deviation compared with FedPop, which further highlights the tuning stability of FedPop.

## 4.2 VALIDATION ON REAL-WORLD CROSS-SILO FEDERATED SYSTEMS

As described in Section 2, previous hyperparameter tuning algorithms focused on small-scale benchmarks and simple model architectures. To indicate the effectiveness of FedPop on real-world FL applications, we further conduct experiments on three large-scale benchmarks: (1) PACS (Li et al., 2017), which includes images that belong to 7 classes from 4 domains Art-Painting, Cartoon, Photo, and Sketch. (2) OfficeHome (Venkateswara et al., 2017), which contains 65 different real-world objects in 4 styles: Art, Clipart, Product, and Real. (3) DomainNet (Peng et al., 2019), which is collected under 6 different data sources: Clipart, Infograph, Painting, Quickdraw, Real, and Sketch. All images are reshaped with larger sizes, i.e., 224x224. Following the setting proposed by Li et al. (2021), we apply cross-silo (Li et al., 2020) FL settings and assume each client contains data from one of the sources (domains), but there exist feature distributions shift across different clients. We use a more complex network architecture, i.e., ResNet-18, as the backbone. We set the tuning budget $(R_t, R_c)$ to $(1000, 200)$. More details about the settings are provided in Appendix.

In Table 2, we report the evaluation results of the target model after tuning by SHA or its combination with FedEx or FedPop. We highlight the performance improvements achieved by the proposed method compared with the competitors, where FedPop surpasses the others up to 2.72% and indicates smaller accuracy deviations. These results indicate the effectiveness of FedPop on real-world FL scenarios with a smaller number of clients, large-scale private datasets, and more complex network architectures.

Table 2: Evaluation results of different hyperparameter tuning algorithms on three real-world cross-silo FL benchmarks with feature distribution shifts.

| Tuning Algorithm | PACS | OfficeHome | DomainNet |
|---|---|---|---|
| SHA | $68.71_{\pm7.38}$ | $38.65_{\pm14.82}$ | $71.41_{\pm6.56}$ |
| | $(76.53_{\pm12.54})$ | $(57.64_{\pm12.21})$ | $(79.41_{\pm11.81})$ |
| FedEx | $73.47_{\pm3.06}$ | $42.99_{\pm8.72}$ | $71.68_{\pm6.13}$ |
| | $(80.61_{\pm5.68})$ | $(58.40_{\pm10.77})$ | $(78.96_{\pm10.71})$ |
| FedPop | $\mathbf{75.17}_{\pm1.18}$ | $\mathbf{45.71}_{\pm7.64}$ | $\mathbf{73.59}_{\pm3.58}$ |
| | $(\mathbf{85.37}_{\pm2.12})$ | $(\mathbf{62.76}_{\pm7.38})$ | $(\mathbf{81.78}_{\pm3.14})$ |

Table 3: Ablation study for different components in FedPop on CIFAR-10 benchmark.

| Tuning Algorithm | CIFAR-10 | | |
|---|---|---|---|
| | IID | Non-IID $(Dir_{1.0})$ | Non-IID $(Dir_{0.5})$ |
| SHA | $72.08_{\pm2.52}$ | $52.41_{\pm12.47}$ | $53.47_{\pm8.53}$ |
| | $(72.12_{\pm3.48})$ | $(40.75_{\pm10.63})$ | $(34.56_{\pm7.10})$ |
| FedPop-G | $74.91_{\pm3.08}$ | $68.41_{\pm5.47}$ | $61.14_{\pm5.45}$ |
| | $(72.74_{\pm2.99})$ | $(62.37_{\pm7.62})$ | $(51.13_{\pm14.07})$ |
| FedPop-L | $74.24_{\pm2.52}$ | $71.50_{\pm1.87}$ | $64.43_{\pm2.86}$ |
| | $(71.54_{\pm3.28})$ | $(64.40_{\pm3.37})$ | $(53.36_{\pm8.48})$ |
| FedPop | $\mathbf{76.69}_{\pm1.02}$ | $\mathbf{73.50}_{\pm0.31}$ | $\mathbf{69.99}_{\pm0.42}$ |
| | $(\mathbf{74.49}_{\pm0.56})$ | $(\mathbf{66.44}_{\pm2.67})$ | $(\mathbf{57.31}_{\pm3.02})$ |

## 4.3 ABLATION STUDY

To illustrate the importance of different FedPop components, we conduct an ablation study on CIFAR-10 benchmark considering *IID* and *Non-IID* settings. The results are shown in Table 3. We

first notice that applying only one population-based tuning algorithm, i.e., either `FedPop-L` or `FedPoP-G`, already leads to distinct performance improvements on the baselines, especially when the client's data are *Non-IID*. Moreover, employing both functions together significantly improves the tuning results, which demonstrates their complementarity.

### 4.4 CONVERGENCE ANALYSIS ON NON-IID IMAGENET-1K

To further demonstrate the scalability of `FedPop`, we display the convergence analysis of `FedPop` on *full-sized* ImageNet-1K, where we distribute the data among 100 clients in a *Non-IID* manner. Hereby, we set $(R_t, R_c) = (5000, 1000)$ and report the average local testing results of the active clients after communication round $r$. We provide more details about the experimental setup in Appendix.

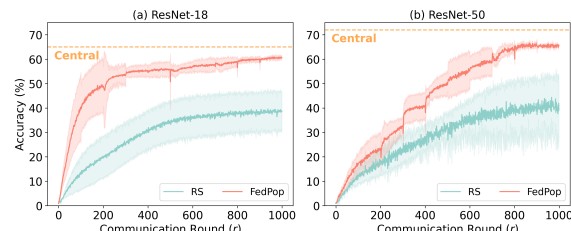

Figure 5: Convergence analysis on *Non-IID* ImageNet.

As shown in Figure 5, we discover that `FedPop` already outperforms `RS` from the initial phase, indicating its promising convergence rate. Besides, we also observe a reduced performance variation in `FedPop`, which further substantiates the benefits of evolutionary updates in stabilizing the overall tuning procedure. Most importantly, `FedPop` achieves comparable results with centralized training, indicating its scalability for large-scale FL applications.

### 4.5 COMPARISON UNDER DIFFERENT SYSTEM DESIGNS

In this section, we analyze the tuning methods under different system designs. Hereby, we demonstrate the effectiveness of `FedPop` with different tuning budgets. To adapt the tuning process according to different $R_t$, we consider 2 possibilities of resource allocations: (1) Varying the number of tuning processes $N_c$ from $\{5, 10, 15, 20\}$ and fixing the per process tuning budget $R_c$ to 400 rounds (200 for FEMNIST). (2) Varying $R_c$ and fixing $N_c$ to 5 (10 for FEMNIST). Here, we select $R_c$ from $\{200, 400, 800, 1600\}$ ($\{100, 200, 300, 400\}$ for FEMNIST).

We report the results in Figure 6. First, we observe that `FedPop` outperforms both `FedEx` and the baseline `RS` in all

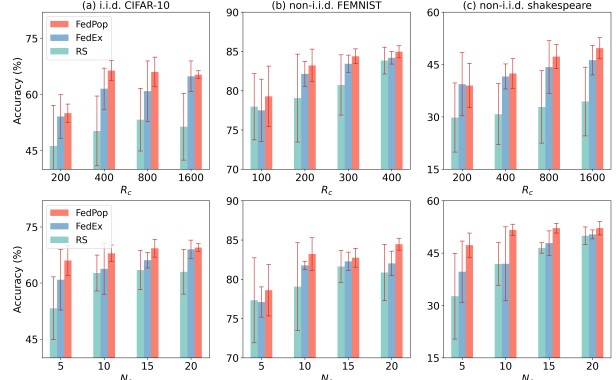

Figure 6: Evaluation results with varying the communication budget for each configuration $R_c$ (top), and varying the number of tuning processes $N_c$ (bottom).

experimental setups, indicating its robustness against different system designs. Also, we observe that a larger communication budget per process $R_c$ leads to better tuning results, while initializing more tuning processes (larger $N_c$) does not lead to obvious performance improvement. This reveals the importance of having a sufficient tuning budget for each configuration.

## 5 CONCLUSION AND OUTLOOKS

In this work, we present a novel population-based algorithm for tuning the hyperparameters used in distributed federated systems. The proposed algorithm `FedPop` method performs evolutionary updates for the hyperparameters based on the member performance among the population. Its global component `FedPop-G`, is applicable for tuning hyperparameters used in server aggregation and client local updates, while for a fine-grained tuning of hyperparameters for clients updates, we apply the fine-grained `FedPop-L`. `FedPop` achieves state-of-the-art results on three common FL benchmarks involving IID or Non-IID data distributions. Moreover, its superb validation results on real-world FL with feature distribution shifts, as well as on distributed Non-IID ImageNet-1K, demonstrate its effectiveness and scalability of FL to more complex applications.

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
