# OpenReview forum: "FedPop: Federated Population-based Hyperparameter Tuning"
_ICLR.cc/2024/Conference — ICLR 2024 Conference Withdrawn Submission_

### Official Review · Reviewer_wxrg · 2023-10-26

**Soundness:** 4 excellent
**Presentation:** 4 excellent
**Contribution:** 3 good
**Rating:** 8
**Confidence:** 4

**Summary:**

The paper proposes the use of evolutionary algorithms to finetune hyperparameters on the fly in a federated learning setting. The resulting algorithm, FedPop, consists of two nested evolutionary algorithms: one local, optimizing the hyperparameters used for local updates at the clients, and another global, concerning the hyperparameters of the server update. This nested optimization strategy paired with the mutations introduced by the evolutionary algorithm substantially increases the number of hyperparameters FedPoP explores during training at virtually no extra computational cost. That translates into significant improvements in performance, as demonstrated in experiments on a number of common benchmarks for Federated Learning.

**Strengths:**

Hyperparameter optimization in federated learning is a relevant and challenging problem, and the application of evolutionary algorithms in this context is indeed novel, to the best of my knowledge. The main ideas of the paper are well motivated and supported by strong and extensive empirical results in a number of relevant benchmarks. Moreover, the paper is very well written and easy to read.

**Weaknesses:**

While the main idea of the paper is novel, I do not think it is particularly innovative, since this seems to be a straightforward application of evolutionary algorithms to hyperparameter optimization. The experimental evaluation, albeit extensive and well designed, is not entirely clear in a few points, as I pointed out in the questions below.

Minor points:
- In “Afterwards, we randomly sample addition K HP-vectors”, I believe the authors meant to say “additional”.
- In Section 3.4.4., the sentence “which saves the communication and computation costs” reads as if FedPop incurred none of these costs, which is probably not the intended meaning. Maybe “saves in communication […]” or “reduces the communication […]” would read better.

**Questions:**

1. In the related work section, the following sentence is not clear “Wang et al. (2019) tunes only the local optimization epochs based on the client’s resources”. What exactly is being optimized here and how?
2. In Table 1, the distinction between global and finetuned models is not entirely clear from the text. Do I understand it correctly that the finetuned version includes one last update over the global model using local data?
3. According to Algorithm 1, FedPop returns $N_c$ models. Do the authors report the average over these $N_c$ models, or the best model among them?
4. In the experiments, it is not always clear what the baselines are.
    - Since RS or SHA are used as a starting point for FedEX and FedProp, are the results for SHA in Tables 2 and 3 obtained with twice the tuning budget? In contrast, what does “None” mean in Table 1?
    - In Figure 5, are FedPoP and RS started from the same population?

---

> ### Author Response · Authors · 2023-11-11
> **Thanks for the reviews!**
>
> We highly appreciate the reviewer for spending time reviewing our work and providing constructive feedback to us. We address the reviewer's comments and concerns as follows:
>
> ### Weakness ###
> Minor points: We will modify our presentation following the suggestions from the reviewer.
>
> ### Questions ###
> Q1: [1] optimizes the update steps of the active clients by estimating the resource consumptions of different clients.
> Q2: Yes, the *finetuned* model is the *global* model optimized with an additional local step at each client using the optimized HPs.
> Q3: We report the performance of the best model among $N_c$ models for all tuning methods.
> Q4: The impression that we are using *RS* or *SHA* as the starting points for *FedPop* stems from our misleading wording "population constructor". The only difference between *FedPop*, wrapped with *RS* or *SHA*, is that we eliminate part of worse-performing HP-configurations in *SHA*. All experiments (*RS*, *FedPop+RS*, *FedEx+RS*), shown in the Table, start from the same population. We will refine our presentation in the revised version for better clarity.
>
> [1] Wang S, Tuor T, Salonidis T, et al. Adaptive federated learning in resource constrained edge computing systems[J]. IEEE journal on selected areas in communications, 2019, 37(6): 1205-1221.

---

### Official Review · Reviewer_KeXM · 2023-10-29

**Soundness:** 2 fair
**Presentation:** 2 fair
**Contribution:** 2 fair
**Rating:** 5
**Confidence:** 4

**Summary:**

This paper proposes a HP-tuning algorithm, called Federated Population-based Hyperparameter Tuning (FedPop), as an online ”tuning-while-training” framework, offering computational efficiency and enabling the exploration of a broader HP search space. FedPop employs population-based evolutionary algorithms to optimize the HPs, which accommodates various HP types at both the client and server sides. Specifically, FedPop employs population-based evolutionary algorithms to optimize the HPs, which accommodates various HP types at both the client and server sides.

**Strengths:**

1. The tuning-while-training framework is important for FL due to its high efficiency.
2. Experiment result are done for real-world cross-silo FL applications.

**Weaknesses:**

1. No explanation about why RS initializes $N_c = R_t / R_c$.
2. Obviously, the average validation loss of all active clients, is dynamic with the convergence for each HP-configuration. Decoupling the effect between convergence and the HP-configuration is important to implement tuning-while-training. Smaller validation loss may come from convergence instead of good HP-configuration. However, the paper does not consider this problem. Thus, it is hard to believe the proposed method can actually tune HP well.
3. The baseline is weak, RS and SHA seem to be simple methods in tuning hyper-parameters. Maybe a greedy search algorithm can be better than RS.
4. In table 1, the accuracies of IID and NonIID cases are too small. For CIFAR-10, the centralized training can obtain > 93% accuracy. And the dir=1.0 and 0.5 are not very strong data heterogeneity. Many works can obtain higher accuracy of this. Could authors explain more about this?

**Questions:**

See weaknesses.

---

> ### Author Response · Authors · 2023-11-11
> **Thanks for the reviews!**
>
> We highly appreciate the reviewer for spending time reviewing our work and providing constructive feedback to us. We address the reviewer's comments and concerns as follows:
>
> W1: $N_c$ (total HP-configurations) and $R_t$ (total communication budgets) are pre-defined before the tuning process starts, according to them we compute $R_c$ (total budgets for each HP-configuration).
> W2: We note that our HP-configurations are tuned simultaneously (as shown in the "In Parallel" in our algorithm description). We always compare the scores of the HP-configuration at the same tuning stage, i.e., after the same number of communication rounds.
> W3: *FedEx* is a strong and concurrent competitor. Also *SHA* applies a greedy algorithm to eliminate the worse-performing HP-configurations in the intermediate steps.
> W4: We note that the results in our experiments are executed with CIFAR10 distributed among 500 clients in total following [1], while the existing works usually apply less clients.
>
> [1] Khodak M, Tu R, Li T, et al. Federated hyperparameter tuning: Challenges, baselines, and connections to weight-sharing[J]. Advances in Neural Information Processing Systems, 2021, 34: 19184-19197.

---

### Official Review · Reviewer_T6YN · 2023-10-31

**Soundness:** 2 fair
**Presentation:** 2 fair
**Contribution:** 2 fair
**Rating:** 3
**Confidence:** 4

**Summary:**

This paper focuses on the hyperparameter optimization or HPO problem in the federated learning (FL) setting. Building upon the FL+HPO setting of Khodak et al. (2021), where the weights from models corresponding to different hyperparameters (HPs) can be aggregated (the "weight-sharing" assumption from Khodak, et al. (2021)), the paper proposes the Fedpop algorithm that optimizes both the server and client HPs. For both sets of HPs, Fedpop utilizes a population-based HPO scheme. The algorithm simultaneously initiates multiple HPO processes from random configurations. Within each process, the client HPs are initialized within a local region around the initial configuration, and then evolved via a population based training scheme with the set of clients forming the population, where weak client HP candidates are replaced by perturbations of strong HP candidates, effectively performing a local search around the initial point. Periodically, both the server and client HPs are synchronized, and HPs for HPO processes with weak performance are replaced with perturbed version of HPs for HPO processes with strong performance. Empirical evaluation of Fedpop highlights  gains over the existing Fedex scheme in IID and non-IID per-client data distribution setting and also in the cross-silo setting. The paper also presents the performance of the proposed Fedpop under various FL setting.

**Strengths:**

The proposed Fedpop algorithm is able to handle both server and client side HPs, and does not require a final federated model training, further extending what we can do with HPO in the FL setting without relatively high communication overhead.

The empirical evaluation utilizes various problem setups such as IID and non-IID per-client distributions, a large-scale cross-silo setup and a large-scale ImageNet-1k training. The performance of the proposed Fedpop is extremely favorable compared to the baselines.

This paper proposes an interesting combination of local searches for client HPs and global search for all HPs, leveraging population based training at different granularities. This is probably the key to the strong empirical performance.

**Weaknesses:**

To the best of my understanding, Fedpop is only applicable to optimizer related hyperparameters (learning rate, momentum, etc) and not to architectural hyperparameters (such as number of layers or activation choices) much like Fedex as it relies on the "weight-sharing" assumption, and thus is limited in scope. Section 2, page 2 (end of paragraph on "Hyperparameter Tuning for FL System") mentions "it does not impose any restriction on [...] model architecture". But it seems that the architecture is pre-selected and not-tuned, and only optimization related HPs are tuned. The Fed-Opt setup depicted in Figure 1 and discussed in Section 3.1 assumes that the weights $\mathbf{w}^k$ corresponding to different client HPs $\alpha_i, \beta_i^k$ can be aggregated. This limits the scope of the HPO problem considered. With Fedex, one can conceptually utilize the outer SHA loop to optimize more general HPs (server and client side), but that is not possible with Fedpop.


Other specific points:
- There appear to be some notational issues in equation (3) and the ensuing discussion. It is possible that I am misunderstanding the presentation, but there might not be a need for $H_j$, and authors can directly use $h_j \sim U(\cdots)$ everywhere. Alternately, the authors can present the equation is a more unambiguous manner.
- It is not clear to me if Fedpop requires each client to simultaneously maintain multiple model weights for the $N_c$ HPO processes, and if the aggregator ever needs to simultaneously maintain $N_c$ (or even $N_c \times R_c$) weights. If that is the case, it would seem that the per-client and aggregator memory overhead is significantly high (especially if we are trying to tune large models).

**Questions:**

- If I understand Flora correctly, it only requires a single FL training, which is done once the hyperparameters have been selected. There is no federated "retraining" in Flora -- just a single federated training in the whole process. Also it is not clear what is meant by "client's local HPs". This is true for Fedex, but Flora handles global hyperparameters (such as architectures and such). What am I missing here? Can this be corrected or clarified?

- What is $N_c$ and how is it related to $R_c$ and $R_t$? In Section 3.1, $N_c$ is defined as the number of clients, while in Section 3.3 $N_c$ is the number of initial HP configurations, and thus, the number of HPT processes. $K$ is used as the number of active clients, but also as the number of perturbed client HPs within the $\Delta$-ball around $\beta_i^0$ of the $i$-th tuning process. Can this be clarified?

- It is not clear what kind of parity is maintained between Fedex and Fedpop. Are all algorithms allowed a total of $R_t$ number of communication rounds? There is no $R_t$ in Fedpop, and the number of communication rounds in fedpop naively seems to be $2 N_c R_c$ (the $R_c$ loop, and the $N_c$ loop, and one round of communication for Fedpop-L and one round for Agg). So is $N_c = R_t / (2 R_c)$? How is the corresponding $N_c$ (the number of initial configurations) in SHA for Fedex selected? It would seem that this would significantly affect Fedex performance.

- Fedex does not really modify the server side hyperparameters $\alpha$ (but starts with multiple to then have the outer SHA loop). How much of the improvement of Fedpop over Fedex attributable to the fact that the $\alpha \setminus \beta$ hyperparameters are being tuned in Fedpop and Fedex is only given 5 configurations of $\alpha$?  There is a  version of Fedex which utilizes an outer SHA (as used in the empirical evaluation) which allows it to try way more server+client HPs. So it seems a bit odd to see that Fedex only tries 5 $\alpha$ values (in the table in Figure 2), which means SHA is seeded with only 5 configurations, which is a bit odd. One would assume that SHA would start with a larger number of configurations.

- How is SHA used for Fedpop? SHA usually utilizes RS to initialize the search, and progressively eliminates candidates. In Fedpop, the elimination (in Fedpop-L and Fedpop-G) are handled internally. So what is SHA doing for Fedpop?

- "Tuning-while-training" is the basis of various popular schemes such as Fedex, as well as, more traditional HPO schemes such as Successive Halving and Hyperband. "Tuning-while-training" is also the basis of the (well-studied) bilevel hyperparameter optimization problem. However, "tuning-while-training" heavily relies on the "weight-sharing" assumption (that is, weights from models with different HPs can be aggregated). Is Fedpop enabling "tuning-while-training" in a new setup?

- How does FedPop address C1 and C2? Retraining (C1) is only avoidable with the "weight-sharing" assumption. How does Fedpop handle/mitigate the mentioned evaluation bias with non-IID data? The discussion in 3.4.4 appears to be without any evidence. Zhou et al. (2021) performs a single FL training on a single HP configuration so, in the notation of this paper, its communication overhead is at most $R_c$. It is not clear why gradient-based HPO from Khodak et al. (2021) is more expensive (in terms of computation or communication). It is also not clear why conducting Fedpop-G every $T_g$ rounds & performing local searches "mitigates the noise" introduced by the non-IID-ness of the problem. These appear to be strong claims without sufficient evidence. Can the author provide evidence for these claims?

- When the local and global HPs are changing throughout the optimization, and $\alpha^\star, \beta^\star$ are the final best HPs selected by the Fedpop algorithm, what is the guarantee that any of the per-HPO-process weights $\lbrace \mathbf w_{i} \rbrace_{i=1}^{N_c}$  (or per-client depending on what $N_c$ denotes) returned by the Algorithm 1 is the same as if a single FL training was initiated from scratch with fixed HPs $\alpha^\star, \beta^\star$ and a max communication budget of $R_c$? With RS or SHA, it is ensured that the model corresponding to the selected HPs is trained for $R_c$ (the max allocated communication budget per HP configuration).  With a bilevel formulation (considered in various bilevel federated HPO such as in FedNest, SimFBO [A] or FedBiO [B]), this is guaranteed if convergence is guaranteed. Retraining from scratch also ensures this but is computationally expensive. Without any such guarantee, it is possible to claim that there is no need for a "retraining" but the final selected model is only trained partially, making the no-retraining claim somewhat limited in scope.

> - [A] Yang, Yifan, Peiyao Xiao, and Kaiyi Ji. "SimFBO: Towards Simple, Flexible and Communication-efficient Federated Bilevel Learning." NeurIPS (2023).
> - [B] Li, Junyi, Feihu Huang, and Heng Huang. "Communication-Efficient Federated Bilevel Optimization with Local and Global Lower Level Problems." NeurIPS (2023).

-  The **Evo** update (as presented) appears to be not related to its input $\mathbf{h}$? It seems like we randomly sample a quantity, and then perturbing it irrespective of the input $\mathbf{h}$. Is that the correct understanding?

- (super minor) HPO seems to be a common acronym for hyperparameter optimization/tuning. Is there any particular reason authors decided to utilize the HPT acronym instead?

---

> ### Author Response · Authors · 2023-11-11
> **Thanks for the reviews!**
>
> We highly appreciate the reviewer for spending time reviewing our work and providing constructive feedback to us. We address the reviewer's comments and concerns as follows:
>
> ### Weakness
> W1: In this work, we focus on the optimizer-related HPs, but FedPop can also be applied to architectural HPs, since our perturbation algorithm **Evo** can also be applied for architectural HPs (e.g., adding or deleting the network layers).
> W2.1: We thank the suggestions from the reviewer and will refine our notation part accordingly.
> W2.2: *FedPop* does not require retaining any model checkpoints at any client, the storage usage of FedPop is the same as naive FedAvg.
>
> ### Questions
> Q1: In FLoRA, there is retraining of the whole FL system required (as shown in Line 8 of Algorithm 1 from their paper). "client's local HPs" means the HPs that are used in the client's local update.
> Q2: The notation in Section 3.1 for $N_c$ is a mistake. $N_c$ means the number of initial HP-configurations. $K$ is the number of active clients. We thank the reviewer for the point and will refine our notations in the revised version.
> Q3: Yes, all methods allow a total number of $R_t$ communication rounds. In our experiments for *FedPop*, $R_t = N_c \times R_c$ and there is no additional round for the $Agg$ process, i.e., an execution of $Agg+Loc$ is considered as one round. We will refine our notation in the revised version.
> Q4: We refer the reviewer to our ablation study in Table 3, where we indicate the effectiveness of *FedPop-G* (tuning both $\alpha$ and $\beta$). The values shown in Figure 2 are the values for *FedEx* and *FedPop* wrapped with *RS*. We will reformulate our presentation to improve the clarity.
> Q5: The introduction of SHA is only to eliminate a specific fraction of the worse-performing tuning process (HP-configurations) in the intermediate steps to save the overall tuning budgets.
> Q6: The main purpose of our "tuning-while-training" procedure is to eliminate the need for costly retraining after the hyperparameter optimization.
> Q7: To address the non-IID problem, we store the historical performance of the hyperparameter configurations and compute a weighted mean of the "history" (*FedPop-G*). We also assume that the non-IID-ness can be mitigated in the local HP tuning when we update the client local HPs based on the results of $K$ active clients (*FedPop-L*).
> Q8: Like the method proposed in [1], our goal is not to find out the best HPs but to optimize a model that performs well after exhausting all tuning budgets.
> Q9: The **Evo** function itself is not related to its input, it only perturbs the value of the HPs.
> Q10: We appreciate the suggestion and will modify this accordingly.
>
> [1] Jaderberg M, Dalibard V, Osindero S, et al. Population based training of neural networks[J]. arXiv preprint arXiv:1711.09846, 2017.

---

### Official Review · Reviewer_RYh9 · 2023-10-31

**Soundness:** 2 fair
**Presentation:** 2 fair
**Contribution:** 4 excellent
**Rating:** 3
**Confidence:** 4

**Summary:**

The authors propose FedPOP, which in spirit is the extension of Jaderberg et. al., 2017 to the federated setting. The authors show how to apply population based HP tuning to the typical server and client-side hyperparameters that appear in federated learning. They show the benefit of their method in a range of empirical studies.

**Strengths:**

The paper is well-motivated and strikes me as a well-executed extension of Jaderberg et al 2017 to the federated setting. The problem of federated hyperparameter tuning is highly relevant - especially as in cross-device FL we cannot assume for the training-data and setup to be repeatable across different runs. Making maximal use of parallel tuning processes through population based evolution is a good approach. I especially enjoy the consideration of iid vs. non-iid and discussion surrounding evaluation - as well as the proposed solution through validation score decay. This is a unique problem that arises in FL and I appreciate the authors addressing it.
The number of different dataset considered is highly appreciated.

**Weaknesses:**

The paper describes an algorithm with a lot of hyper-parameters, training & evaluation settings with a lot of specified details across different dataset. As a reader, it is hard to keep an overview of the exact settings, assumptions and baselines as well as the choice of those hyperparameters (e.g. number of clients, dirichlet sampling prob, R_c, R_t, N_c, search-space, annealing rates and much more). I would highly appreciate a detailed table of required parameters to reproduce experiments and a clear definition of how baselines are compared (e.g. is R_t equalized) in the Appendix. Some of my below questions stem from a lack of overview.

It took my quite some time to understand the definition of components and the relationship $R_t=N_c*R_c$ from reading the text. The code-base conveniently uses the same help-string "Training epochs for server" for all three elements ;). A short and concise definition of these at the beginning of 3.3 would have been helpful. You seem to be equating "training rounds", "communication budget" and "maximum resources" in your exposition, although in any other FL-context, these terms can have very different meaning.

Some statements and results seem to be contradicting (e.g. are weights initialized from the optimal RS/SHA setting vs. learning-curves, see below). At this point, the paper requires some additional care to clarify how empirical studies have been done.

I believe this paper to have very high potential and I will raise my score should the authors alleviate my concerns about the existing empirical evaluations and clean up their exposition. Should the authors further expand their experiments with a discussion about the evolution on hyper-parameters and an empirical discussion around the power-law method for validation scores, I will consider raising my score further.

Thanks for putting this paper together, I enjoyed digging into it.

**Questions:**

- Algorithm 1 considers $K$ active clients per round. However these K same clients seem to be utilized by all tuning processes. What is the assumption here? Do we assume that the same K clients participate in $N_c$ different "parallel" training runs, executing one local training configuration after the other - or do we assume that at any point in time, there are $N_c*K$ clients available to participate in a specific training run?
- RS or SHA as initial population construction. I understand that you pick the best configuration $\alpha, \beta, w$ from RS or SHA as starting point $\alpha_i, \beta_i, w_i$ for all tuning processes $i$. Consequently, for example Table 1 "None" rows stand for the starting-point of the subsequently applied Tuning Algorithms. FedPop (and FedEx) therefore enjoyed an additional budget of $R_t$ communication rounds, beyond what the "naive" tuning algorithms have been granted. What is therefore missing is a baseline where RS and SHA alone have been granted the additional budget to find better alternatives, i.e. through longer convergence times or additional random configurations. While I believe that you show that population-based training does provide additional benefits compared to naive RS, Table 1 (and the other results) is not equalized by resources-used, which is what you claim to base your experiments on.
- In the context of the previous question, I would have expected the learning curves of Figure 5 and Appendix Figure 4 to look different: Why is FedPop not starting from the best-configuration of RS in terms of evaluation accuracy? These curves seem to suggest that you start from random $w$, which is contrary to your exposition in 3.4.
- If my understanding is correct, then the communication and computation for the population construction with RS or SHA is not accounted for in the stated budget for FedPop. Could you please provide the configuration for creating these initial starting points?
- One of the most interesting insights in Jaderberg et. al 2017 was to see the "history" of hyperparameters that lead to the final model. For example, it would 'discover' learning rate warmup and annealing. I would expect it to be highly insightful to see the same analysis for these federated experiments. Could you provide that for some of your runs?
- The problem of federated evaluation is generally quite challenging, since not all clients are assumed to be available for evaluation all the time. I would love to understand the consequence of this for your experiments. Concretely: Assuming we do have access to the entire validation set at the server (which corresponds to standard FL evaluation for C10 and related "toy" datasets), how much standard-deviation do we observe by randomly sampling $K$ active clients for evaluation around the "true" global dataset-evaluation? How much does the choice between a) single-sample estimate $s_i$ per tuning process b) power-law decayed estimate $s_i$ and c) the oracle whole-federation $s_i$ for the computation of $FedPop-G$ influence the selection of hyper-parameters as well as the final result? This ties in to my previous question: Do you use the same $K$ clients' validation sets for computation of $s_i$ across all tuning processes $i$ or do you additional stochasticity comparing across different $K$ validation sets. While I intuitively understand why you propose the power-law method, I see a lack of empirical evidence to measure its benefits.
- Hyperparameters are missing. E.g. how many clients do you consider for the individual experiments? I see some settings in the code, but the paper should be complete for reproducibility. Unless you are reproducing more hyperparameters from previous work (4.1.1) than is apparent from the text, in which case please be more explicit.
- I have assumed, but realize now you haven't stated explicitly: Did you equalize $R_t$ between FedEx and FedPop? Beyond that, comparing Table 1 SHA FedEx with Table 4 FedPop-G, I believe we can see what happens if you use FedPop to only optimize the server-side parameters, which would put it onto equal-footing with FedEx in terms of what the methods can optimize. If the resource budget is equalized between those two, it would be valuable to point out that FedPop outperforms FedEx even when constraining FedPop to optimizing server-side h-params only, assuming we fix the local parameters to the optima found through SHA.
- I do not understand Algorithm 1 in the appendix. My understanding of "Using RS or SHA as population constructor" follows 3.4, meaning we find the initial $\alpha, \beta, w$ through either RS or SHA. Algorithm 1 seems to perform SHA interleaved with FedPop. Please clarify.

---

> ### Author Response · Authors · 2023-11-11
> **Thanks for the reviews!**
>
> We highly appreciate the reviewer for spending time reviewing our work and providing constructive feedback to us. We address the reviewer's comments and concerns as follows:
>
> ### Weakness ###
> We appreciate the important point raised by the reviewer and will add an explanation table for the different parameters and values used in our experiment in the Appendix. Additionally, we will make better clarifications for the terms: "training epochs for server", "training rounds", "communication budget", and "maximum resources" in the revised version. Finally, we will add discussions and visualizations for the evolution of HPs in the revised version. Thanks again for the valuable advice and compliments!
>
> ### Questions ###
> Q1: At each communication round, each tuning process (in total $N_c$ different tuning processes) obtains $K$ different active clients.
> Q2,Q3,Q4,Q9: We assume the impression that we are applying *FedPop* based on the outcomes of *RS* or *SHA* stems from our misleading wording "population constructor". We will refine our wording for better clarity. Concretely, all tuning methods use the same tuning budget, i.e., the same total communication rounds are executed. Besides, all tuning methods start from randomly initialized HPs as well as models. The only difference between *RS* or *SHA* as a population constructor (or to be better formulated as a wrapper) only lies in "whether to terminate some of the tuning processes in the intermediate steps" in *FedPop* and *FedEx*.
> Q5: We will add visualizations for the evolution of HPs following the suggestions in the revised Appendix.
> Q6: We will add more analysis and experiments on the effectiveness of our power-law decays strategy for the estimation of the evaluation scores.
> Q7: We will add more experimental details for all datasets in the revised version.
> Q8: The tuning budget for FedEx and FedPop are the same (same $R_t$).

---

### Official Review · Reviewer_5toX · 2023-11-03

**Soundness:** 2 fair
**Presentation:** 2 fair
**Contribution:** 2 fair
**Rating:** 5
**Confidence:** 4

**Summary:**

This paper presents a hyperparameter optimization (HPO) algorithm for federated learning (FL). Compared to existing works, the proposed is more efficient since it employs an online "tuning-while-training" framework. The evolutionary algorithm is used to search for the optimized hyperparameters during the FL training process. Empirical results show that the proposed FedPop algorithm outperforms random search (RS) and successive halving (SHA) baselines.

**Strengths:**

1. The problem tackled in this paper is well-motivated. Reducing the computational efficiency is an important and difficult problem for the HPO of FL since each run of the FL process is expensive.

2. The organization of this paper is good. The proposed method is clearly described and easy to follow.

**Weaknesses:**

1. RS and SHA are two very simple baselines that are not enough to demonstrate the significance of the proposed method. Some other SOTA methods such as Hyperband and BOHB should be compared. In Section 4.4, the convergence analysis is only done over RS and FedPop. Why are the learning curves of other baselines in Table 1 not shown in Figure 5?

2. Some claims of this paper are not well supported. For example,

- The authors highlighted several times that FedPop can be conducted in a parallel and asynchronous manner. However, it's not clear which steps can be asynchronously executed and how it could affect the efficiency of the HPO process.

- In Section 3.3, it is mentioned that "the elimination might also discard HP-configurations which lead to promising results but perform poorly at early stages". However, this issue seems to also exist in the proposed FedPop due to the replacing steps in FedPop-L and FedPop-G. Can this issue be eliminated or alleviated in the proposed method?

3. The lack of theoretical analysis reduces the significance of the proposed method. Due to the "tuning-while-training" strategy, the hyperparameters used in one FL run keep changing. Whether the FL process can converge under such dynamic hyperparameters becomes an essential question. Even though the learning curves of two experiments are shown in Figure 5, it's not enough to verify the convergence property of FedPop in other general cases. The authors have also noticed that "too distinct HP-vectors for the active clients would lead to unstable performance", which implies that the change of hyperparameters could affect the convergence of FL in some scenarios. Some theoretical analysis would help to address this concern.

**Questions:**

1. In Algorithm 1, each $w_i^k$ is trained using different hyper-parameters (e.g., different batch_size). Then, can they be fused directly?

2. The evaluation bias (C2) is considered in FedPop-G but not in the FedPop-L algorithm. However, for the non-IID case, the good hyperparameters for one client may not be suitable for other clients. Can you provide some analysis on the effectiveness of FedPop-L in the non-IID case?

3. What aggregation algorithm did you use in the experiments? Does the proposed method work well for various FL aggregation strategies (e.g., FedAvg, FedProx, etc.)?

4. In Table 1, why is the performance of some fine-tuned models worse than that of the global model?

---

> ### Author Response · Authors · 2023-11-11
> **Thanks for the reviews!**
>
> We highly appreciate the reviewer for spending time reviewing our work and providing constructive feedback to us. We address the reviewer's comments and concerns as follows:
>
> ### Weakness ###
> W1: We select the same competitors following [1]. Additionally, we compare with *FedEx*, which is a sophisticated method compared with *RS* and *SHA*. We assume that Hyperband requires running multiple times of *SHA* (multiple brackets) to achieve promising results, which would drastically increase the tuning time for FL applications. Also, BOHB is not suitable for FL applications since it would require costly retraining of the overall system after obtaining the best HPs. We will add the curves for *FedEx* in the revised version in Figure 5.
> W2.1: The tuning process can be executed asynchronously (the second for-loop in Algorithm 1). We will describe this in more detail in the revised version.
> W2.2: To address the possible problem caused by early-stopping, we apply a resampling of HPs from their original distribution with a specific probability value in **Evo**.
> W3: We thank the reviewer for the suggestion and will consider adding this in the revised version.
>
> ### Questions ###
> Q1: We assume that the local models trained with different HPs can be fused directly, which is also done in [1].
> Q2: We refer the reviewer to our ablation study in Table 3. In *FedPop-L*, we use *K* active clients as our population, which gives us a more informative evaluation of the HPs compared with using only a single client.
> Q3: We use *FedAvg* in our experiments, we will add an experiment combining *FedProx* with different tuning methods in the revised Appendix.
> Q4: We assume that sometimes, applying finetuning would lead to the already promising global model overfits to the small client local datasets.
>
> [1] Khodak M, Tu R, Li T, et al. Federated hyperparameter tuning: Challenges, baselines, and connections to weight-sharing[J]. Advances in Neural Information Processing Systems, 2021, 34: 19184-19197.